# Wireless Sensor Networks for Enabling Smart Production Lines in Industry 4.0

**Brecht De Beelde *** , **David Plets** and **Wout Joseph**

Department of Information Technology, Ghent University/IMEC, 9052 Ghent, Belgium;
David.Plets@UGent.be (D.P.); Wout.Joseph@UGent.be (W.J.)
* Correspondence: Brecht.DeBeelde@UGent.be

**Featured Application: Design of hybrid wireless sensor networks for industrial environments, used for connecting sensor modules that capture information, enabling automated production lines and AI-driven assembly.**

**Abstract:** With the deployment of data-driven assembly and production factories, challenges arise in sensor data acquisition and gathering. Different wireless technologies are currently used for transferring data, each with different advantages and constraints. In this paper, we present a hybrid network architecture for providing Quality of Service (QoS) in an industrial environment where guaranteed minimal data rates and maximal latency are of utmost importance for controlling devices and processes. The location of the access points (APs) is determined during the initial network-planning action, together with physical parameters such as frequency, transmit power, and modulation and coding schemes. Instead of performing network-planning just once before the network rollout, the network is monitored continuously by adding telemetry data to the frame header of all data streams, and the network is automatically reconfigured in real-time if the requirements are not met. By not using maximum transmit powers during the initial roll-out, more APs are needed, but coverage is guaranteed when new obstructions such as metallic racks or machinery are added. It is found that decreasing the transmit power by 6 dB gives the best trade-off between the number of required APs and network robustness. The proposed architecture is validated via simulations and via a proof-of-concept setup.

**Keywords:** network-planning; sensor networks; wireless communication; IIoT; PHY layer; MAC layer; FoF; Industry 4.0

## 1. Introduction

### 1.1. Background

Industrial warehouses, factories, and manufacturing processes are rapidly evolving towards digitized systems [1]. Technologies such as big data analysis, Internet of Things, and artificial intelligence (AI) are part of the factories-of-the-future (FoF) paradigm and are key to the Industry 4.0 (I4.0) concept. Due to the increased transfer of information, reliable communication systems are fundamental to the success of FoF. In traditional factories, wired communication networks are created and configured during the installation of the factory. This does not allow for flexibility, and finding and repairing a broken cable is a time-consuming and costly task. Wireless communication not only has the advantage of reduced wiring costs, but it also enables faster factory turnarounds, such as when a production line changes. Furthermore, it adds support for mobile systems and is easier to add redundancy.

Currently, a multitude of wireless technologies exists for industrial Internet-of-Things (IIoT) applications. Examples at sub-GHz frequency 868 MHz are low-power wide-area networks (LPWAN), such as LoRa, Z-wave, and DASH7. LPWAN for IIoT applications is

discussed in [2], and the use of LoRa for I4.0 applications is discussed in [3]. In the 2.4 GHz industrial-scientific-medical (ISM) band, Bluetooth Low Energy (BLE), IEEE 802.15.4 (Zig-Bee, Thread, 6LowPAN), and IEEE 802.11 (WiFi) are widely used. New wireless technologies such as software-defined networks (SDN) improve the overall system's efficiency and have shown to facilitate I4.0 applications [4]. With the existence of different technologies, the first choice to be made when designing a wireless network for IIoT is what technology will be used. This decision can be made on technological aspects, such as battery life, coverage, and throughput, but also on the technologies that are available on specific sensor modules that are needed. One example of the need for high-throughput communication in FoF is quality inspection via computer vision in an electric motor production hall [5]. Next to the high-throughput wireless local area network (WLAN), wireless sensor networks (WSN) with strict latency requirements are often used for process and robot control. Timing is also critical for the communication of automated guided vehicles with a backend server. Body area networks are a third category, as monitoring and protecting workers' health is another key challenge of FoF [6]. This can be obtained from radar-based monitoring systems or via information gathered by wearables. An example of the use of wearables in an industrial environment is posture tracking of factory operators, which not only allows for optimizing workbench configurations, but also identifying workload identification [7]. As different wireless technologies are used for different purposes, it is clear that the main challenge for network operators is the interoperability and coexistence of existing technologies as well as integration with state-of-the-art technologies, such as the software-defined radio (SDR).

When the choice of which wireless technology to use is made, a network topology is selected and the location of access points (APs) is determined during the network-planning phase. Traditionally, network-planning and rollout is a one-time action during which the locations and configurations of APs are determined [8]. This assumes a static environment where a new planning action is performed whenever the environment changes. In FoF, requirements are dynamic, machinery is added or replaced, and the floor plan might be reorganized from time to time [9]. Therefore, maintenance is important after the network rollout, as an updated environment layout can cause coverage holes. Additionally, firmware and security updates may be required, and interference issues may arise when a wireless technology is added. An industrial environment is often characterized by the presence of large metallic objects [10]. Hence, layout changes in the environment can cause network issues that arise long after the network rollout.

Different I4.0 categories exist, such as IIoT, big data analytics, fog computing, and robotics, and it is clear that communication and exchange of information are critical [11–13]. The availability of sensor data allows big data analysis [14,15], data-driven production reconfiguration [16], and technologies such as AI and machine learning algorithms [17] and fault detection [18], amongst others.

### 1.2. Related Work

Haile et al. presented a data-driven network-planning framework for outdoor fifth-generation (5G) network-planning [19]. network-planning for WLAN by maximizing network efficiency is presented by Bosio et al. [20]. For WiFi network-planning, multiple commercial tools exist as well, including solutions from Ekahau and Netspot.

Pramudianto et al. proposed a service-oriented architecture for monitoring and controlling the network [21]. This architecture allows for automatic modification of the network parameters and physical (PHY) settings when the network performance degrades. For the PHY layer, only the frequency is changed when interference is detected. The dynamic channel allocation was also discussed by Khaleel et al. [22]. Mao et al. presented an iterative algorithm to obtain optimal transmit powers, beamforming settings, and MAC-layer settings, such as time allocation [12,23].

SDN allows for more flexibility and scalability [4]. It is also used to monitor the network by keeping track of all device states. Govindaraj et al. proposed a reallocation

of resources to ensure zero downtime of the factory by over-dimensioning the network and using SDN to reconfigure the network [24]. Lin et al. provided a network-planning algorithm for SDN [25]. A network planner for multi-hop mesh networks in an industrial environment was presented by Chen et al. [26]. The planner was tested via simulations. Gong et al. described a genetic algorithm for large-scale planning in industrial environments using over-dimensioning [9]. Other technologies and protocols were proposed for IIoT, such as the reference architecture model for I4.0 by De Melo et al. [27], but these have the main drawback that they are not backward-compatible with legacy devices.

### 1.3. Novelty

State-of-the-art solutions and commercial network-planning tools only support a single wireless technology and perform open-loop networking planning, that is they define AP locations based on a predefined floorplan and perform an on-site measurement campaign. Tools that use a monitoring system adjust a single parameter, that is, a monitor network interference, and automatically switch channels based on channel occupancy.

Instead of only monitoring interference and modifying channel settings, we present a closed-loop algorithm for monitoring the packet error rate (PER), received signal strength, signal-to-noise, and latency, and reconfiguring the PHY and medium access control (MAC) parameters of the network: modulation and coding scheme (MCS), output power, bandwidth, and frequency settings. The closed-loop algorithm allows to quickly determine and solve network issues, and enables wireless machine-to-machine communication with guaranteed quality of service (QoS) in an industrial environment. The framework is not only tested via simulations, but experimental validations in an industrial lab are also performed.

To the best of the authors' knowledge, this is the first closed-loop framework that simultaneously adjusts PHY and MAC layer parameters of multiple wireless technologies, including WiFi, ZigBee, LoRa, and SDR, and has been implemented and experimentally tested in an industrial environment using IEEE 802.11 and IEEE 802.15.4 technologies. This closed-loop framework ensures a fully automated system with limited manual maintenance, while providing QoS without a fully redundant network architecture. This makes it unnecessary to perform an on-site measurement campaign prior to network-planning, as the network automatically adjusts to the environment. Furthermore, the support for legacy and new wireless technologies makes the framework suitable for the design of networks for FoF, where different types of technologies are required.

In Section 2, we first present the overall system architecture and then describe link monitoring on a per-packet basis, PHY network-planning, QoS monitoring, and PHY layer configuration. Section 3 presents the simulation and validation scenarios. In Section 4, we present and discuss simulation and validation results of the proposed framework. Section 5 concludes this paper.

## 2. Methodology

In this section, we present the closed-loop framework that monitors the network status and automatically reconfigures the network to meet certain QoS requirements. We start in Section 2.1 with a presentation of the considered industrial environments. In Section 2.2 we provide an overview of the overall system architecture. As monitoring data play a crucial role, we present a telemetry system in Section 2.3. We describe the initial physical layer network-planning for heterogeneous networks in Section 2.4 and then discuss the calibration of the coverage map based on monitoring data in Section 2.5. Section 2.6 presents the monitoring loop and reconfiguration algorithm.

### 2.1. Considered Environments

In the literature, the envisioned environments for FoF and I4.0 are production and assembly halls [5,28], manufacturing and production facilities [1,3,4,21,24], wood and metal processing facilities [10], and industrial warehouses [2,29]. The considered environments in this paper are factory warehouses, processing facilities, and assembly halls. These

environments are characterized by large room dimensions and the presence of multiple large objects that can block radio frequency (RF) communication. Because of the room dimensions, it cannot be covered by a single AP or gateway, and multiple APs are needed.

### 2.2. System Architecture

The overall system architecture is shown in Figure 1. We consider a hybrid wireless network in which multiple technologies with different levels of customization co-exist. We categorize all devices into either the access point (AP) or client, irrespective of the used technology (Client and access-point device types originate from WiFi terminology). In IEEE 802.15.4 ZigBee, the client is called the *end device* and the access point is called the *coordinator*. In Bluetooth Low Energy (BLE), the terms *peripheral* and *central* are used. In LoRa, the terms *end device* and *gateway* are used. For simplicity, we use the terminology access point and client for all technologies.). Both AP and client contain an RF transceiver device and we categorize the transceivers according to the wireless technology that they support. Moreover, one physical device can implement multiple RF transceivers and act as an AP for one technology and as a client for another technology. APs are connected to the intranet via wired connections, or via a gateway using another technology. An example of the latter is a ZigBee gateway that acts as an IEEE 802.15.4 AP and as an IEEE 802.11 client, that is, data from the ZigBee clients are gathered and sent to another AP via WiFi. This is shown in Figure 1, where the APs for Technology 2 at the right of the floor plan acting as clients for Technology 1, and connect to the intranet via AP1.

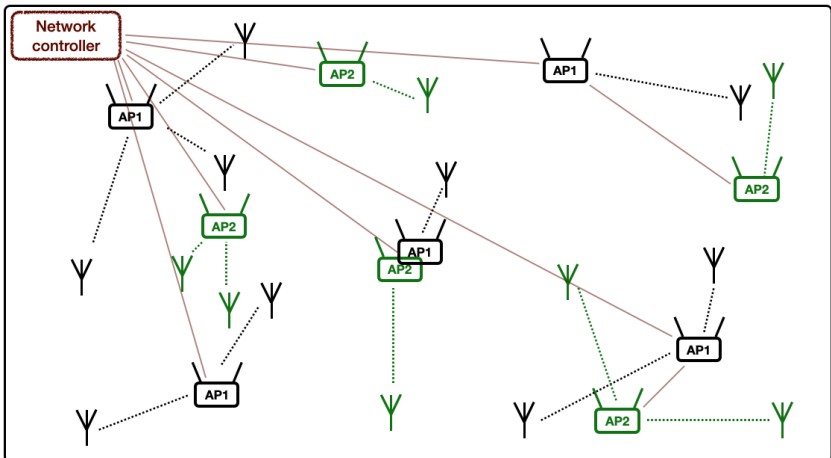

**Figure 1.** Overall system architecture with multiple wireless technologies. Clients wirelessly connect to an access point, and access points are connected to the network controller via a wired connection, or via another (1-hop) access point. Wired connections are represented by a straight line, whereas wireless connections are represented by a dotted line.

Wireless communication links are categorized according to the required QoS, that is, maximum delay and latency, and minimum throughput. Jitter and packet loss are QoS parameters that relate to MAC-layer network configurations or are caused by interference. Packet loss can also be caused by RF blockage, which we handle from a throughput perspective.

During the initial network-planning, APs are placed on the floor plan based on an initial set of requirements and based on an initial floor plan layout. In this work, we monitor the link quality on a per-packet basis. Once the networks are operational and data are being sent, monitoring data are used to calibrate the simulated parameters, such as received signal strength and throughput. In a continuous monitoring process, a centralized local network controller can reconfigure PHY and MAC layer configuration settings to allow an adaptive system that can be used in a dynamic environment. Indeed, the floor plan layout of the factory can change, for example, due to adjusted assembly lines or added machinery, but the application requirements may also change. The network controller is similar to the gateway that is proposed in [30], that is, it keeps track of the current state

of the network, can provide local processing with low latency, and can also send data to the cloud. The different phases, from initial network-planning to the monitoring loop, are visualized in Figure 2.

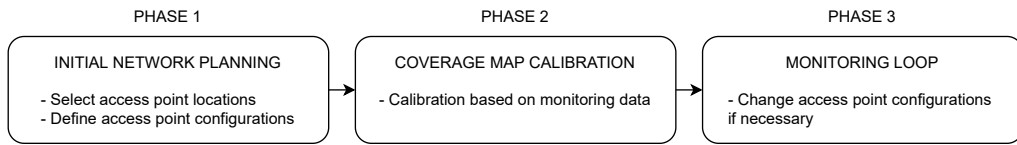

**Figure 2.** Different phases, from initial network-planning to the monitoring loop.

### 2.3. Telemetry Data

Having accurate and timely monitoring data is crucial for our closed-loop network framework. On the other hand, the monitoring data should introduce limited overhead. Different methodologies exist to obtain telemetry data in order to know the state of the network. The most frequently used method for link monitoring is extracting received signal strength (RSS) information from the data streams. An implementation of a control system using telemetry via received signal strength indicator (RSSI) readings is provided in [31]. Commercial monitoring systems also exist for monitoring network link quality via specific sensor modules that use the RSSI information from the packet headers.

In-band network telemetry (INT) is a monitoring system which, compared to network monitoring by traffic probing and equipment polling, uses the data streams in the network for acquiring monitoring information by adding telemetry data on a per-packet basis. INT-based monitoring methods are common for wired networks with programmable data plane technology [32–35], and research on using INT-based monitoring for wireless networks is ongoing [36,37]. On the MAC layer, an INT header is added that contains information on the network state, that is, downlink RSSI, throughput, latency, and packet error rate information. This information is used to determine whether QoS requirements are met, and to reconfigure the network if they are not met.

### 2.4. Initial Physical Layer Network Planning

During the initial PHY network-planning phase, that is, step 1 in Figure 2, the location and configuration of the APs is defined. As the determination of AP locations is a non-deterministic polynomial (NP) problem, we use a heuristic placement algorithm that defines the optimal location where the APs need to be placed to provide full coverage. We extend an existing network-planning tool [28] which implements the placement algorithm by using a statistical PL model of the environment. The number of required devices depends on the environment, on the used transceiver type (and their receiver sensitivity), on the transmit (TX) power setting, and on the required data rate.

Before starting the initial network-planning, we defined the geometry of the floor plan and the characteristics of the objects that are present in the room. We model the available transceiver devices based on PHY parameters, for example, by the radio type and frequency band in which they are operational. One AP can contain multiple transceiver radios, for example, both a WiFi and ZigBee RF chipset.

#### 2.4.1. Coverage and Throughput

The heuristic is based on the statistical *PL* model from (1), which is validated for industrial environments [28]. In (1), $d$ is the distance in m, $PL_0$ is the reference *PL* at distance $d_0$ of 1 m, $n$ is the dimensionless *PL* exponent, $L_R$ is the rack attenuation in dB, and $\chi_\sigma$ is a fading term in dB based on a normal distribution with zero mean and standard deviation $\sigma$. For the considered environments, we use a $PL_0$ value of 46.91 dB and *PL* exponent value $n$ of 1.96 [28]. The standard deviation is 2.39 dB and the rack attenuation $L_R$ is assumed to be 4.6 dB. As a PL model depends on the specific environment, a one-

day measurement campaign is proposed in [29] to obtain a specific *PL* model for the environment where network-planning is needed.

$$PL(d) = PL_0 + 10\,n\,\log_{10}(d/d_0) + \sum_i L_{R_i} + \chi_\sigma \tag{1}$$

The maximum theoretic throughput depends on *PL* via the used modulation and coding scheme (MCS). In general, a more complex MCS can be used for a higher received power, which results in a higher data rate. The received power depends on the TX power and *PL*, and as the maximum TX power is limited, *PL* defines the coverage and throughput. The network-planning tool can also be used to calculate theoretic throughputs based on a floor plan geometry and AP configurations, by calculating received power at every grid point.

### 2.4.2. Delay and Latency

Apart from coverage and minimum throughput, maximum delay and latency are important QoS parameters for IIoT. Delay is mainly caused by the processing time in the network equipment, and by higher-layer protocols. In the literature, MAC layer enhancements are proposed to decrease end-to-end latency in multi-hop networks [38]. Examples are an airtime reservation system [39], slotted MAC [40], and minimization of duty cycling delays [41].

Khanafer et al. proposed to minimize latency via a cluster-based WSN design [42]. In this work, we use the hybrid concept that is proposed by Shrestha et al. [43], where a hybrid network consisting of WiFi and ZigBee devices is used to enable a WSN in a train. The network is arranged in clusters, and ZigBee technology is used within clusters. As a multi-hop ZigBee network has high end-to-end delays and traffic congestion, the intra-cluster links use WiFi technology. However, the train environment in [43] differs significantly from the industrial environment we consider in this paper. We assume that WiFi APs, which are typically connected close to the ceiling, are connected to the network controller via Ethernet. As this might not always be the case for BLE and ZigBee APs, they can connect to a gateway that relays information via WiFi. MAC-layer time-slicing is used to obtain low-latency communication links.

If the latency exceeds a given threshold, it is assumed that too many clients are connected to a single AP, and it is verified whether clients can connect to a nearby AP, by performing a throughput calculation of the network with the constrained AP removed from the floor plan.

### 2.4.3. Other Cost Functions

During the initial network-planning step, other constraints can be taken into account. Examples are minimization cost functions, for example, the minimal cabling cost [28], or the minimum power consumption of the clients. A possible constraint is that APs can only be placed on walls, for example, when no cable trays are present near the ceiling.

### 2.4.4. Hybrid Network Planning

We extend the network-planning tool from [28] to enable network-planning of hybrid networks, for example, a network consisting of WiFi and IEEE 802.15.4 devices, in which every technology has its coverage requirements. The technology layer with the more stringent requirements is optimized first. The initial network-planning action results in the location of the APs together with the PHY settings, such as frequency, TX power, and MCS. Based on propagation models and floor plan settings, a coverage map is calculated containing received power and the maximum achievable download speed based on the possible MCS at every grid point.

For the initial network-planning, we differentiate between legacy technologies using off-the-shelf transceivers and technologies for which we can reconfigure PHY parameters. Technologies for which we cannot modify PHY parameters via the network controller will

use PHY settings that provide the best possible coverage, such as the maximum allowable TX power. Technologies for which we can control PHY parameters from the network controller will be configured in such a way that we have headroom to solve coverage issues that arise after the network has been rolled out. This results in an increased number of APs that are required to get full coverage of the environment. The impact of the additionally required APs is discussed in Section 4.2.1.

### 2.5. Coverage Map Calibration

From the initial network-planning result, we have a coverage map with theoretically received power at every location. The coverage map is based on a statistical PL model. Therefore, the calculated received power can differ from actual received power, for example, due to different material characteristics resulting in different attenuation, or when the floor plan is not fully accurate. From the monitoring system discussed in Section 2.3, we gather measurement data such as latency and RSSI from the client devices. It is also expected that reported RSSI values show an offset compared to the received power $P_{RX}$, as the RSSI depends on the actual chipset.

To be able to make decisions based on the reported RSSI values in the monitoring loop, a calibration of the calculated coverage map is needed. This calibration is performed for each receiver and is visualized as step 2 in Figure 2. Figure 3 shows the algorithm for the coverage map calibration algorithm.

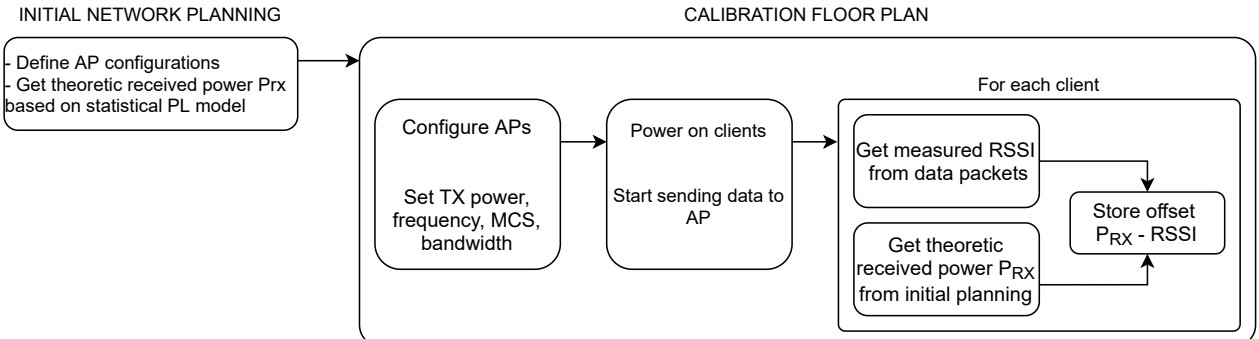

**Figure 3.** Coverage map calibration algorithm: storing offset between measured received signal strength information (RSSI) and simulated received power ($P_{RX}$) for every client.

After the initial network-planning, the APs are installed in the factory hall and the network is set up. When the clients start sending and receiving data, the mean value of a number of RSSI samples of the data packets is calculated and compared to the theoretically received power value from the initial network-planning. For each client, we store the offset between $P_{RX}$ and RSSI and we will take this offset into account when making decisions based on RSSI.

Note that the location of the clients is assumed to be known, which is an acceptable assumption due to the extensive localization methods that are available for indoor environments [44], including visible light positioning [45–49] or the time difference of arrival techniques [50]. Localization based on RSSI is also possible [51] and can be integrated into the network-planning framework.

### 2.6. Closed-Loop Monitoring

After the calibration of the coverage map, the network controller continuously monitors INT data and verifies that the coverage requirements are met, that is, the latency is smaller than threshold $L_{max}$, the packet error rate is below threshold $PER_{max}$, and the measured RSSI of all devices is above the threshold $RSSI_{min}$ that is needed to obtain a certain MCS. The threshold $RSSI_{min}$ depends on the throughput requirements. A PHY reconfiguration is triggered when the QoS requirements are not met. This monitoring loop is visualized as step 3 in Figure 2 and presented in Figure 4.

MONITORING LOOP

**Figure 4.** Monitoring algorithm: monitor packet error rate (PER), received signal strength information (RSSI), signal-to-noise ratio (SNR), and latency (L), and a PHY reconfiguration is triggered when QoS requirements are not met.

For each client, we obtained the available QoS parameters. If the latency exceeded threshold $L_{max}$, we calculated the received power at every location assuming that the AP to which the client is connected is not present. If clients that are connected to the constrained AP have the required QoS when connecting to another AP, for example, the throughput is guaranteed via another AP, they will reconnect to decrease the load on the constrained AP. We measure signal-to-noise ratio (SNR) and RSSI on the client devices so we can calculate the noise floor (NF). When the NF exceeds the threshold $NF_{max}$, we evaluate the available frequency channels for the AP and client devices and obtain the frequency occupation of the complete network. We then select the channel that is least used. If the measured RSSI is higher than the required threshold, an MCS can be used that enables a data rate higher than the required throughput. If the measured RSSI is higher than the threshold but the packet error rate exceeds threshold $PER_{max}$, another frequency is selected to minimize interference. If the measured RSSI of all clients connected to one AP is considerably higher than the minimum to receive power corresponding to the required throughput, the TX power of the AP can be lowered. If the RSSI of a client is lower than the threshold, for example, as the result of added machinery or an updated floor plan, the required throughput is not guaranteed and therefore the network has to be reconfigured. We added a virtual obstruction to the floor plan in the center between the client and AP locations with an attenuation equal to the difference between calibrated and measured RSSI, and then reconfigured the PHY parameters of the AP.

The reconfiguration can consist of updated frequency settings, updated power settings, and an updated MCS or bandwidth mode, depending on the technologies that are used. For the network reconfiguration, the placement of APs is taken as a constraint to reconfigure the network without changing the AP locations, so that this reconfiguration can be done without human interaction.

Whenever a reconfiguration is triggered for one specific AP, a separate process is started on the network controller that verifies whether other APs can also be reconfigured. Furthermore, we verify whether a lower MCS could be sufficient for all connected clients. If this is the case, the TX power is decreased. As this reconfiguration is a time-consuming task, it runs in a separate process and is re-initiated whenever a new reconfiguration is triggered.

### 3. Algorithm Implementation and Validation via Simulations and Measurements

In the previous section, we presented the generic methodology of the closed-loop network-planning and monitoring framework. In this section, we first present the implementation of this framework in Section 3.1. The framework is tested via simulations and validated by measurements in an industrial lab. The simulation settings are presented in Section 3.2, and the validation measurements are presented in Section 3.3.

*3.1. Implementation*

3.1.1. Overview

A multitude of wireless technologies exist, and the choice of which technology to use is often dictated by the availability of certain sensors. For the implementation of the algorithm, we consider the following wireless technologies. As WLAN technologies, offering large data rates, we use IEEE 802.11n and 802.11ax WiFi, both operational in the 2.4 and 5 GHz frequency bands. Possible use cases of WLAN networks in an industrial environment are video monitoring and remote maintenance. WLAN technology is also present in tablet devices used by workers, for example, to download specific user manuals or register data while performing maintenance. For WSN technologies, we use LoRa and IEEE 802.15.4 ZigBee to transmit low-throughput data from various sensors. Apart from standard ZigBee, we also consider the SDR introduced in [52], which is an IEEE 802.15.4 compliant transceiver with flexible MAC and PHY layers, of which we can modify the channel bandwidth (BW). The characteristics of the SDR are presented in Table 1.

**Table 1.** IEEE 802.15.4 PHY software-defined radio (SDR) characteristics [52].

| Parameter | Standard BW Mode | Narrow BW Mode |
|---|---|---|
| Bandwidth (MHz) | 2 | 0.25 |
| Data rate (kbps) | 250 | 31.25 |
| Sensitivity (dBm) | $-98$ | $-107$ |

As workers can also wear on-body devices, for example, to monitor the worker's health, body area networks are also considered. For body area networks, we use Bluetooth Low Energy (BLE). The on-body devices can communicate to a central worn on-body, for example, a cell phone, or to a fixed off-body AP.

3.1.2. Telemetry

The framework proposed in Section 2 does not depend on a specific monitoring system. We use INT monitoring for the validation of the closed-loop framework, that is, the client adds a header on MAC layer that contains information on the data frames it receives from the AP. Via INT monitoring, we obtain RSSI and SNR from every packet, as well as latency information from a packet stream that is set up via the Click Router framework [53]. In the network controller, we store a moving median of the RSSI, SNR, and latency, using the last 20 data packets. These values are used in the monitoring loop to verify whether QoS requirements are met. The details on the implementation of INT for WiFi and ZigBee devices are provided in [37,54].

3.1.3. Reconfiguration

The PHY parameters that are reconfigurable depend on the technology. The reconfigurable parameters we consider are listed in Table 2 for different technologies.

**Table 2.** Reconfigurable physical (PHY) layer parameters for different technologies.

| Technology | Transmit Power | Frequency | MCS | Bandwidth |
|---|---|---|---|---|
| IEEE 802.11n/ax | ✓ | ✓ | ✓ | |
| LoRa | ✓ | | ✓ | |
| BLE | ✓ | | ✓ | |
| SDR IEEE 802.15.4 | ✓ | ✓ | | ✓ |
| standard IEEE 802.15.4 | ✓ | ✓ | | |

The reconfiguration algorithm is outlined in Figure 5. For IEEE 802.11n/ax WiFi, we can adjust the transmit power, frequency, and MCS. When the throughput QoS is not met and the 5 GHz frequency band is used, we switch to the 2.4 GHz frequency band as PL increases with frequency. When we already use the 2.4 GHz band, we try to lower the MCS if possible. If, by lowering the MCS, we cannot achieve the required throughput, we increase the TX power to the maximum value. For LoRa, we can change the TX power but also change the spreading factor, which defines the used modulation array. As the sensitivity increases with increasing spreading factor, we first try to increase the spreading factor. If this is not possible, given the required data rate, the TX power is increased. For BLE 5.0, we have three different PHY versions: the 1 Mbps PHY used in BLE 4, the 2 Mbps PHY, and the Coded PHY. LE Coded PHY has a lower receiver sensitivity than the LE 1M and LE 2M PHYs. We can switch between the three PHYs, and we can modify TX powers. As BLE uses adaptive frequency hopping, we do not set the frequency channel.

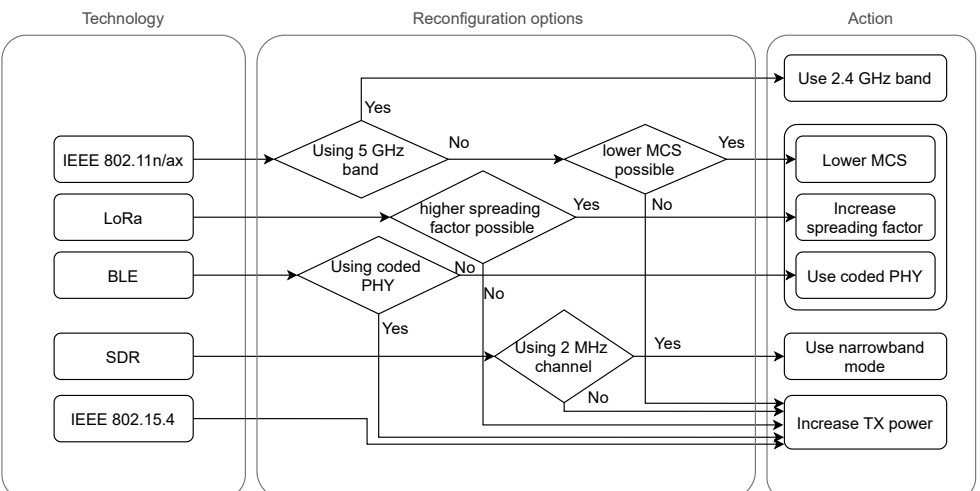

**Figure 5.** Reconfiguration algorithm: reconfiguring PHY parameters for different wireless technologies.

For the chirp spread spectrum (CSS) PHY used by IEEE 802.15.4, we can only change the frequency and TX power. We also consider an SDR which is IEEE 802.15.4 compliant but has a configurable channel BW. By lowering the channel BW from the 20 MHz defined in the IEEE 802.15.4 specification [55] down to 2 MHz, the data rate lowers from 250 kbps to 31.25 kbps and the receiver sensitivity of the radio improves from −97 dBm to −110 dBm. Therefore, using a narrower band, the receiver sensitivity decreases, and the coverage increases at the cost of a lower data rate. If using the narrowband mode is not sufficient, we increase the TX power.

Note that for the technologies listed in Table 2, the maximum TX power is defined by regulations. To have the flexibility for the reconfiguration, we select a default TX power that is lower than the maximum allowable TX power. This results in more APs being placed on the floor plan than necessary, but it allows for more flexible reconfiguration due

to environmental changes. The optimal default TX power is derived from simulations, presented in Section 4.

When a reconfiguration is triggered, a process is started to verify whether settings can be reverted, that is, whether we can decrease TX power. As an example, when machinery is moved in the factory, the RSSI of one link can decrease, causing a reconfiguration. On the other hand, the RSSI of another link can increase so that it is not needed anymore when a certain AP transmits at full power. For WiFi, we also verify whether we can use the 5 GHz frequency band (to minimize interference with other APs). LoRa APs could decrease the spreading factor, and BLE APs that use the coded PHY could revert to the LE 1M PHY.

### 3.2. Simulation

### 3.2.1. Simulation Environment

We test our implementation via simulations for two industrial environments: a factory warehouse with a large number of racks, and a processing facility with larger room dimensions but fewer racks. A floor plan of the two simulation environments is presented in Figure 6, in which the grey objects represent metallic racks. The factory warehouse measures 112 m by 131 m and is part of a truck assembly hall. In this warehouse, 42 metallic racks with a height of 9 m are present, which results in a dense obstruction. The metal processing facility measures 138 m by 175 m, but with only eight racks present, this is a more open environment.

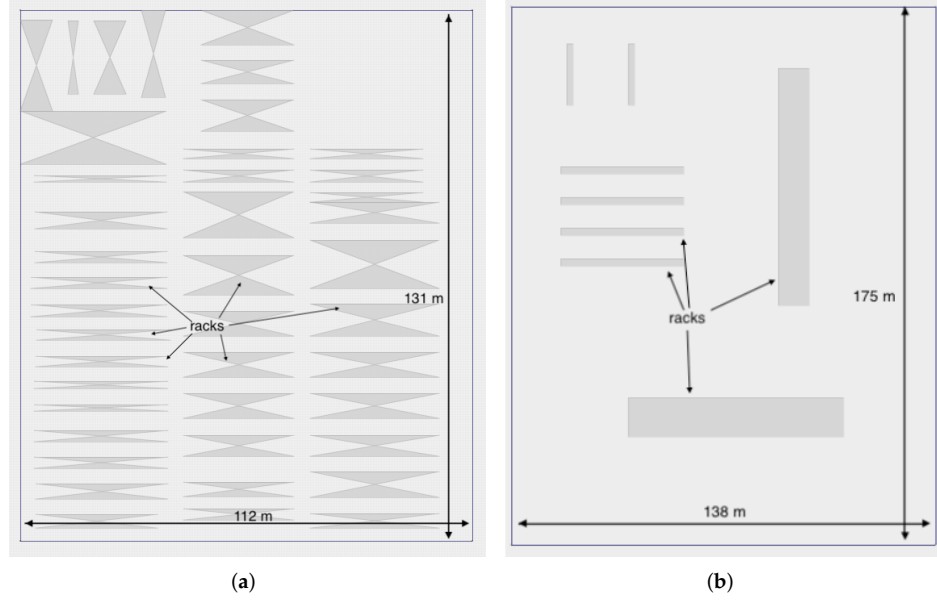

(a)          (b)

**Figure 6.** Floor plan of the simulation environments. Gray objects represent metallic racks. (**a**) Factory warehouse (112 m × 131 m). (**b**) Processing facility (138 m × 175 m).

### 3.2.2. Simulation Settings

For both simulation environments, we deploy a hybrid network consisting of WiFi, ZigBee, and BLE legacy devices, as well as SDR devices with adjustable channel BW. For WiFi, a throughput requirement of 54 Mbps is used, which corresponds to a receiver sensitivity of −68 dBm. For BLE and standard ZigBee, a throughput of 250 kbps is required, whereas for SDR, a data rate of 31 kbps is sufficient, which can be obtained in a narrow BW mode. The receiver sensitivity of standard ZigBee devices is −86 dBm, and for SDR it is −107 dBm. The receiver sensitivity of BLE is −96 dBm.

We compare the required number of APs for both environments and investigate the effect on the number of APs by not using the full TX power. From simulations, we obtain the number of required APs as a function of maximum TX power. We also verify whether

throughput requirements can be met when additional racks are placed on the floor plan after initial network planning.

### 3.3. Validation

#### 3.3.1. Validation Environment

In addition to simulations, we test our implementation via a proof-of-concept (PoC) validation in an IIoT lab measuring 11 m by 30 m and containing a warehouse emulation area with three large centrally placed racks. The floor plan and a picture of the room are shown in Figure 7.

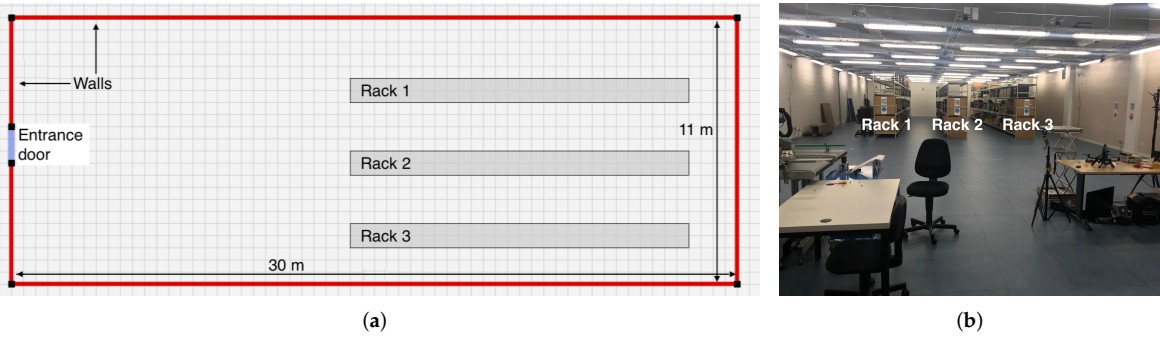

(a)                                                                     (b)

**Figure 7.** Floor plan and picture of the validation environment (30 m × 11 m). (**a**) Floor plan. (**b**) Photograph.

#### 3.3.2. Validation Measurements

Via the PoC measurements, we verify that monitoring data are successfully acquired. Once the APs are placed according to the initial network-planning result, and the networks are operational, multiple data streams are set up between different clients. The monitoring data are used for the calibration of the coverage map, and to trigger a network reconfiguration when the floor plan changes. Modified WiFi and ZigBee APs are used that use INT-based monitoring, and we use the IEEE 802.15.4 SDR [52] for testing the PHY reconfiguration algorithm.

We use a latency threshold of 100 ms. If the link latency exceeds this threshold, the load of the AP is considered too high, and the reconfiguration is triggered. For the clients connected to the AP, it is verified whether other AP exists that can provide the same throughput. We use a packet error rate threshold of 0.1. If the measured packet error rate exceeds this threshold, the AP switches to another channel. As we need the location of the clients to compare the measured RSSI to the calculated received power, the location of the clients is marked on the floor plan.

## 4. Results and Discussion

In this section, we present and discuss the simulation and validation results, starting with initial network-planning in Section 4.1 and simulations of network performance in Section 4.2. In Section 4.3, we present the validation results, and our framework is compared to existing solutions in Section 4.4.

### 4.1. Initial Network Planning

To provide full coverage in the factory warehouse presented in Section 3.2.1, which has an area of 14,730 m$^2$ and contains 42 racks, a total of 23 WiFi APs, 4 ZigBee APs, 2 SDR APs, and 3 BLE APs are required when all APs transmit at maximum TX power. This high number of APs is caused by the presence of a lot of metallic racks. For the processing facility, with a total area of 24,150 m$^2$ but a lower rack count, 17 WiFi APs, 1 ZigBee AP, 1 SDR AP, and 1 BLE AP are required to obtain the throughput rates defined in Section 3.2.2. The total simulation time of initial network-planning for the factory warehouse is 42 min on a laptop with a 2.3 GHz Intel Core i5 central processing unit (CPU).

*4.2. Performance Analysis*

4.2.1. Network Robustness

By not using the maximum TX power, more APs are needed to cover the complete floor plan. This results in an additional installation cost but has the advantage that the network can cope with layout changes by increasing TX powers after the network has been rolled out.

Figure 8 shows the number of required WiFi APs for the two environments as a function of TX power. For the factory warehouse, we need 53 WiFi APs when using a TX power of 15 dBm, and the total number further increases up to 107 when the WiFi APs use a TX power of 10 dBm. Using a TX power of 15 dBm, we need 4 BLE APs, and 7 ZigBee APs. We need 10 SDR APs that have a TX power of 0 dBm. For the processing facility, 56 WiFi APs are required for full coverage when using a TX power of 15 dBm, and this number further increases to 147 when the APs transmit at 10 dBm. Even though the processing facility is much larger than the factory warehouse, a similar number of APs is required.

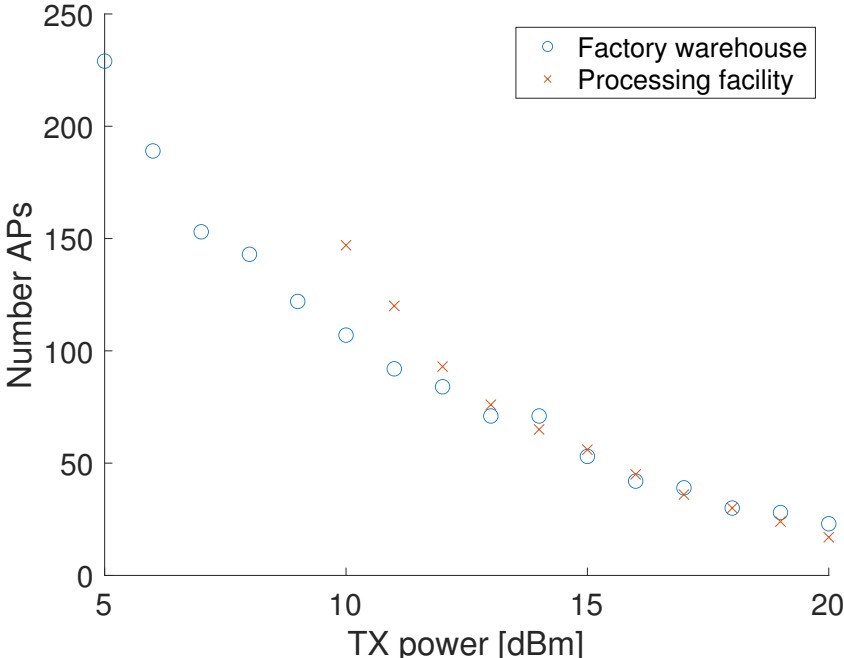

**Figure 8.** Number of required access points as a function of the maximum transmit power of the access points.

To verify the network robustness, we calculate network coverage when racks are added and some of the existing racks are moved. Figure 9 shows the modifications made to the floor plan after the initial network-planning is performed, while Figure 10 shows the network coverage as a function of TX power for different levels of layout changes. Coverage is defined as the number of grid points on the floor plan for which the 54 Mbps throughput requirement is met, divided by the total number of grid points. In this figure, we see that for the factory warehouse environment, coverage is challenged by the presence of the large number of racks when the TX power is below 12 dBm, even when no changes to the floor plan occur. For the processing facility, coverage is above 98.5% for all floor plan modifications, if the TX power is in the range 10 to 14 dBm. For only three rack additions, the coverage stays above 99.6%. For (default) TX powers of 17 dBm or lower, the coverage for the most updated floor plan, that is, 10 added racks, is above 97.5%. A 1 dB increase in TX power results in a coverage of 96.0%. From this figure, the ideal TX power to use during the initial network-planning is 14 dBm.

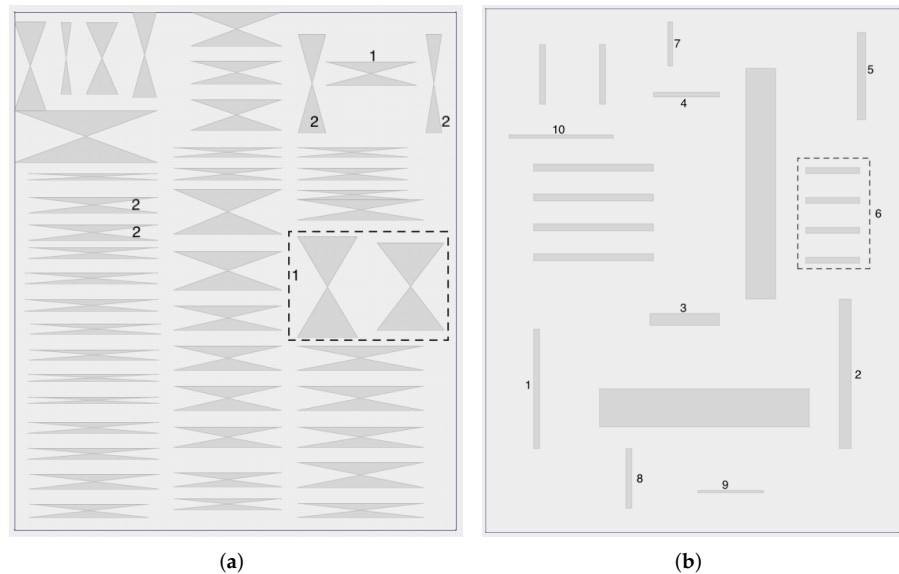

(**a**)  (**b**)

**Figure 9.** Updated floor plan of the simulation environments, with numbers indicating the order in which racks are added or modified. (**a**) Factory warehouse (112 m × 131 m). (**b**) Processing facility (138 m × 175 m).

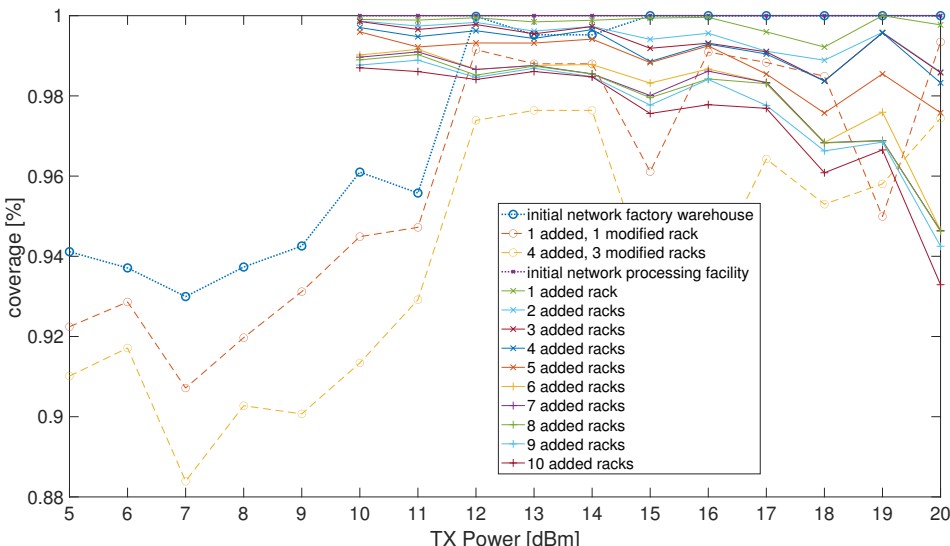

**Figure 10.** Network coverage as a function of the maximum transmit power of the access points.

### 4.2.2. Network Bandwidth

The client's throughput depends on the received power, which depends on PL and TX power. With higher received power, a higher MCS setting can be used. Apart from the maximum throughput of the wireless link, real-life throughput also depends on the bandwidth available at the AP, that is, the uplink data stream is shared by all clients connected to the AP. Furthermore, the number of simultaneously connected clients is often limited.

By using lower TX power, more APs are needed to cover the complete floor plan. However, this also increases the total network capacity. Considering that most commercial WiFi APs have a Gigabit Ethernet link, and can connect up to 20 clients at full download speed, the total network capacity of the factory warehouse increases from 12 Gbps when the APs operate at full TX power, with a total of 460 clients, up to 53 Gbps when the APs use a default TX power of 15 dBm. For the latter, more than 1000 clients can connect.

### 4.3. Validation in a Real-Time Testbed

We validated the setup by deploying a network in the IIoT lab that is presented in Section 3.3.1. We perform network-planning for WiFi and SDR transceivers, which results in the placement of 3 WiFi APs and 1 SDR AP. The floor plan layout is shown in Figure 11, with the WiFi APs represented by black circles, the SDR AP by a green circle, and the clients are represented by hexagons. The lines indicate to which AP the clients are connected. The calculated received power for WiFi clients at every grid point is also shown on the floor plan.

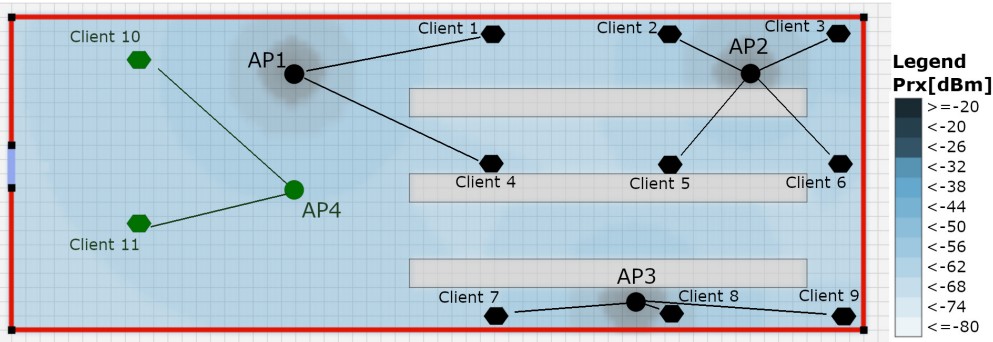

**Figure 11.** Initial network-planning result for WiFi and SDR APs in an IIoT lab measuring 11 m by 30 m. Clients are represented by hexagons and APs are represented by circles. The color indicates the wireless technology (green = SDR, black = WiFi). The lines indicate to which AP the clients are connected.

#### 4.3.1. Calibration

When the network is operational, the clients start streaming and sending data to each other via the APs to which they are connected. Using the INT monitoring framework discussed in Sections 2.3 and 3.1.2, we obtain actual link quality information, which is gathered in the network controller and used to calibrate the coverage map. Table 3 shows the calculated received power and measured RSSI for all clients, for which the locations are indicated in Figure 11.

**Table 3.** Theoretic received power [dBm] and measured RSSI [dBm] for the clients (located at the positions indicated in Figure 11).

| Client | 1 | 2 | 3 | 4 | 5 | 6 | 7 | 8 | 9 | 10 | 11 |
|---|---|---|---|---|---|---|---|---|---|---|---|
| theoretic $P_{RX}$ | −48 | −41 | −42 | −51 | −49 | −49 | −45 | −34 | −48 | −65 | −63 |
| measured RSSI | −56 | −50 | −48 | −57 | −48 | −32 | −36 | −43 | −47 | −82 | −87 |

For clients 1 to 4, all having the same chipset, there is an average offset of −7.25 dBm, that is, the measured RSSI is 6 to 9 dBm lower than the calculated received power. For clients 5 and 6, there is a positive offset, indicating that the rack attenuation that is presumed during initial network-planning was too high. As can be seen in Figure 7b, the racks are not as dense as the racks used in [28]. For clients 7 and 9, there is also a positive offset which cannot be explained based on room geometry but can be due to the chipset of the clients. For the SDR clients, there is a larger offset between theoretically received power and measured RSSI. For every client, the difference between theoretic and measured received power is stored and used as offset during the monitoring loop.

#### 4.3.2. Network Reconfiguration

During the monitoring loop, we artificially increase the processing delay of AP2, located in the upper-right corner of Figure 11, to simulate increased load on the AP. This increases the link latency of the data streams of clients 2, 3, 5, and 6, and causes client 5 to

switch from AP2 to AP3, located close to rack 3 in Figure 11. The number of clients and APs is too limited to cause significant interference.

For the PHY layer reconfiguration, we use an attenuator at the RF input port of the SDR, which we use to gradually decrease the received power. As can be seen in Figure 12a, the communication between nodes 11 and 12 (via AP4) stops when the RSSI drops below −95 dBm when the monitoring loop is disabled. When the monitoring loop is active, a PHY reconfiguration is triggered when the RSSI is below −95 dBm, going from the standard 2 MHz bandwidth channel to the narrow bandwidth mode which has a lower receiver sensitivity. We see in Figure 12b that the reconfiguration takes 10 s, during which no data are sent. After the reconfiguration took place, we could further increase the attenuation until the RSSI reached −110 dBm. When the attenuation is further increased, the transmit power can be increased to maintain the link.

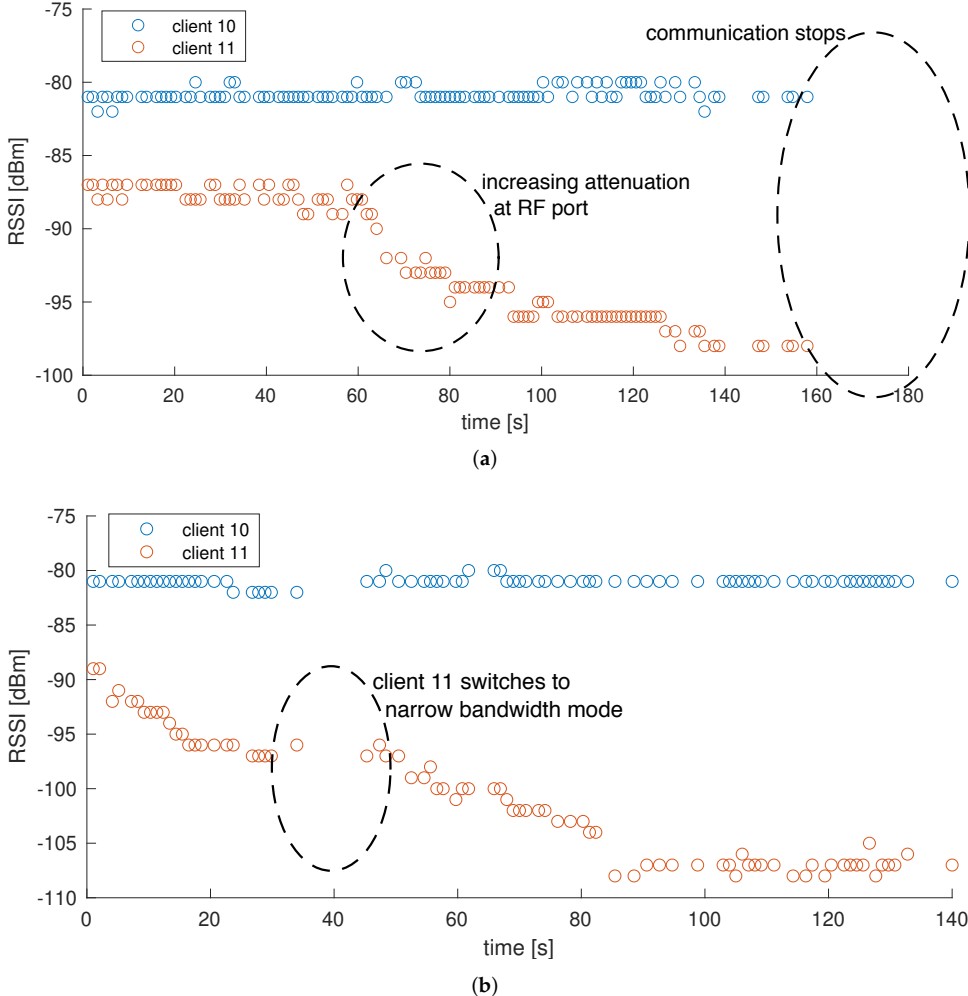

**Figure 12.** Measured received signal strength information (RSSI) from clients 10 and 11, both connected to AP4, after a data stream is set up between the two clients. The attenuation at the RF port of client 11 is gradually increased to test the reconfiguration algorithm. (**a**) monitoring loop disabled. (**b**) monitoring loop active, a reconfiguration is triggered.

### 4.4. Comparison to Existing Network-Planning Solutions

Haile et al. presented a hybrid framework for outdoor network-planning using different radio access technologies [19]. Compared to this framework, we need a more detailed description of the environment and network-planning results, that is, the distances between clients and APs are in the order of meters, rather than km for an outdoor scenario.

On the other hand, the framework from [19] used a fixed user service demand to plan the network, whereas our framework does not need this a priori information.

Compared to commercial WLAN network-planning solutions from Ekahau and Netspot, the main advantage of our framework is that it can be used for more wireless technologies, that is, it is not only WiFi is supported. Furthermore, the commercial network planners perform network-planning during the moment. Troubleshooting by an expert is needed when issues arise after network rollout, whereas the closed-loop monitoring system allows for automatic reconfiguration. The work from Bosio et al. also applies to WiFi technology only, and does not discuss transmission power control techniques, and is tailored on the specific MAC layer options that are available in IEEE 802.11 [20].

The frameworks proposed by Pramudianto et al. [21] and Khaleel et al. [22] only change frequencies when the network performance degrades. When the floor plan changes, or when the load on an AP increases, this will not be sufficient and other settings need to be modified as well.

Chen et al. discussed routing methodologies for a multi-hop network and presented simulation results. However, network-planning does not take into account the propagation characteristics of the environment, and only considers the graph theory. Graph theory is also used by Lin et al. for the network-planning of SDN [25], but a feedback-control loop is developed for dynamic network reconfiguration. The use case, that is, backhaul network-planning, differs and the implementation does not relate to the wireless PHY layer. Govindaraj et al. used SDN in an industrial environment, but envisioned a fully redundant network [24]. Additionally, the heuristic developed by Gong et al. is based on a fully redundant network [9], that is, every client can be served by at least two APs, which results in a much higher number of required APs. By using lower transmit powers in our framework, we also over-dimension the network, but this results in a lower number of total APs which leads to a lower installation cost, and less interference.

In order to have accurate network-planning, a one-day on-site measurement campaign is proposed in [29] to obtain an accurate PL model that is used by the network planner. A one-day measurement campaign and automatic network-planning using the heuristic from [28] already requires less human interaction than performing network-planning via trial-and-error. By using measured RSSI values for the calibration of the calculated coverage map, there is no need for an on-site channel sounding measurement campaign to have accurate attenuation values for the specific objects in the envisioned environment, as the network will adapt to the environment. For any possible environment, we can use a generic PL model from the literature (e.g., [10]) which will automatically adapt to the specific site where the network is deployed based on measurement data.

## 5. Conclusions

In this paper, a closed-loop algorithm is presented that allows for automatic modification of physical network parameters based on actual in situ measurements. After the initial network-planning and configuration, the network is continuously monitored, and physical network settings are optimised based on the measured QoS parameters on the clients. If the received signal strength is too low, TX power of the AP is increased. If the signal-to-noise ratio drops, another frequency is selected, and clients are connected to another AP when the load on one AP is too high. This allows for QoS to be provided without the need to significantly over-dimension the network. Based on simulations, it is found that a TX power of 6 dBm below the maximum TX power gives the best trade-off between coverage performance and the number of required APs. Even if the floor plan is significantly modified after the network has been rolled out, for example, by the addition of metallic racks, QoS is guaranteed. The algorithm is validated via simulations, and experimentally by implementing it in a real testbed. The measurements confirm that the network successfully reconfigures when the RSSI drops, or when the load on one AP increases. Reconfiguring the network based on measurement data eliminates the need for an on-site measurement campaign prior to the initial network-planning in order to fit the path loss model to the

given room. It also allows for automatic maintenance to limit manual interventions in the network configuration.

Because there is no single technology that fits all industrial use cases, multiple coexisting wireless technologies are of utmost importance. The proposed framework performs network-planning for different technologies with different levels of reconfiguration possibilities. This allows not only to perform network-planning in large and complex environments, but it also allows for monitoring of the network and reconfiguring it if necessary. There is support for legacy devices with no configurability, as well as for existing technologies with custom firmware and new technologies, such as SDR. This makes the tool viable for the design of wireless networks of FoF, where support for different wireless technologies is critical, given the multitude of existing sensors already available, and the growing number of new technologies.

Future work includes running a field trial in a larger factory hall with more devices, and analyzing interference. Furthermore, RSSI-based localization of clients will be implemented, and more MAC-layer settings are configured by the network controller, including airtime and packet priorization.

**Author Contributions:** Conceptualization, B.D.B., W.J. and D.P.; methodology, B.D.B.; software, B.D.B. and D.P.; validation, B.D.B.; formal analysis, B.D.B., W.J. and D.P.; investigation, B.D.B.; resources, W.J.; data curation, B.D.B.; writing—original draft preparation, B.D.B.; writing—review and editing, B.D.B., W.J. and D.P.; visualization, B.D.B.; supervision, W.J.; project administration, W.J.; funding acquisition, W.J. All authors have read and agreed to the published version of the manuscript.

**Funding:** This research received no external funding.

**Institutional Review Board Statement:** Not applicable.

**Informed Consent Statement:** Not applicable.

**Data Availability Statement:** Not applicable.

**Acknowledgments:** This work was executed within the imec AAA project Better Than Wired and EOS project multi-service wireless network (MUSE-WINET). The authors would like to thank research group IDLab for the use of the IIoT lab and Patrick Bosch, Jetmir Haxhibeqiri, and Pedro Isolani for their collaboration in the Better Than Wired project.

**Conflicts of Interest:** The authors declare no conflict of interest.

## Abbreviations

The following abbreviations are used in this manuscript:

| | |
|---|---|
| 5G | Fifth Generation |
| AI | Artificial Intelligence |
| AP | Access Point |
| BLE | Bluetooth Low Energy |
| FoF | Factories-of-the-Future |
| I4.0 | Industry 4.0 |
| INT | In-band Network Telemetry |
| IIoT | Industrial Internet of Things |
| ISM | Industrial, Scientific, Medical frequency band |
| LPWAN | Low Power Wide Area Networks |
| MAC | Medium Access Control |
| MCS | Modulation and Coding Scheme |
| MDPI | Multidisciplinary Digital Publishing Institute |
| PHY | Physical layer |
| PoC | Proof-of-concept |
| RF | Radio Frequency |
| SDN | Software Defined Networking |
| SDR | Software Defined Radio |
| PL | Path Loss |

| RF | Radio Frequency |
|---|---|
| QoS | Quality of Service |
| WLAN | Wireless Local Area Network |
| WSN | Wireless Sensor Network |

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
