# Peer review of "Wireless Sensor Networks for Enabling Smart Production Lines in Industry 4.0"

_applsci, doi:10.3390/app112311248_

Round 1
Reviewer 1 Report
The authors present a network architecture that automatically adopts MAC and PHY parameters to optimize an industrial wireless network environment. The presentation of the work is good, but with the major concern that there is no comparison of the work to other works. That makes it really hard to judge the novelty / contribution of the work. For the paper to get a clear contribution - a comparison of the work should be included.
Detailed comments:
- The methods section is large and is a mix of description of test environment and the methods used in the network architectures, think the clarity of the work would improve if these are separated.
Reviewer 2 Report
This paper proposes a hybrid network architecture using Software Defined-Networking and legacy networks 8 for providing QoS in an industrial environment. Overview, this topic of this paper is interesting and this paper is well-organized. However, i have the following comments to improve the quality of this paper
- The authors should add an additional section to overview the related research works, e.g., the literature about wireless sensor networks and Industrial 4.0.
- The authors should add more descriptions to reveal the novelty of this paper, as compared to existing works.
- Please add more descriptions to explain the figure in this paper, such as Fig. 1.
- In order to improve the readability of this paper, the authors should provide a paragraph to reveal the simulation findings at the beginning of Section 3
- Some recent works should be cited in this paper, such as
[1] "Computation Rate Maximization for Intelligent Reflecting Surface Enhanced Wireless Powered Mobile Edge Computing Networks," in IEEE Transactions on Vehicular Technology, vol. 70, no. 10, pp. 10820-10831, Oct. 2021, doi: 10.1109/TVT.2021.3105270.
[2] Energy-efficient cooperative communication and computation for wireless powered mobile-edge Computing[J]. IEEE Systems Journal, 2020: 1-12, doi: 10.1109/JSYST.2020.3020474.
Reviewer 3 Report
I have a number of comments.
1. It is necessary to more carefully analyze the subject area and update the list of references.
2. The abstract is not informative, it is necessary to revise, part of it should be referred to the introduction.
3. Figures are poorly readable and uninformative.
4. Flowcharts of algorithms need to be revised for the presentation of information in each of the blocks.
Round 2
Reviewer 1 Report
Thanks for the revised version of the paper that has improved the readability and quality. However, I think that the work that is used in the added comparison section should be summarized so that it easier to grasp the uniqueness and novelty of the presented work.
Reviewer 2 Report
The authors have addressed all the concerned problems, so i suggest to accept this paper
Author Response
Dear Reviewer,
Thank you for your time reviewing our paper. We appreciate your comments and believe that they have improved the quality of our paper significantly.
Best regards,
Brecht